# OpenReview forum: "INPO: Image-based Negative Preference Optimization for Concept Erasure in Text-to-Image Diffusion Models"
_ICLR.cc/2026/Conference — Submitted to ICLR 2026_

### Official Review · Reviewer_iHuL · 2025-10-20

**Soundness:** 3
**Presentation:** 3
**Contribution:** 3
**Rating:** 4
**Confidence:** 4

**Summary:**

The paper proposed a method to erase concepts from Diffusion based generative models using an adaptation of negative preference optimization. The report results on nudity, objects and style erasure on FLUX.1-Dev.

**Strengths:**

- "Adaptive Erasure Trajectories" in section 4.3 is very interesting and makes the approach easily useable without defining a target concept.
- The results on applying NPO for concept erasure are promising and can be beneficial for future research.
- The paper in general is well written and is easy to follow.

**Weaknesses:**

- The authors talk about "Architecture-dependence" in the introduction as motivation behind their approach but proceed to only show results on FLUX.
- The experiments evaluations are lacking. I'm in particular interested in how the approach benchmarks on SDv1.4 on adversarial prompt datasets (Ring-A-Bell, MMA Diffusion, etc). Especially in comparison with newer approaches such as AdvUnlearn, AGE.
- The paper talks about prompt-dependence but fails to cite or acknowledge [1].
- There are many relevant works that have not been cited. While I am not listing all here, please add relevant works.

[1] Pham, Minh, et al. "Prompt-Agnostic Erasure for Diffusion Models Using Task Vectors."

**Questions:**

- How do you find the concept masks M?
- What is the performance on multi-concept erasure such as on celebrity 100 dataset from MACE [1]?
- Have you re-run red-teaming attacks on FLUX or have you reused prompt sets that were optimized for SDv1.4? I'm surprised to see such low numbers for the baseline nudity generation on FLUX.

[1] Lu, Shilin, et al. "Mace: Mass concept erasure in diffusion models." Proceedings of the IEEE/CVF Conference on Computer Vision and Pattern Recognition. 2024.

---

> ### Author Response · Authors · 2025-11-26
> **Response to Reviewer iHuL**
>
> > **W1: The authors talk about "Architecture-dependence" in the introduction as motivation behind their approach but proceed to only show results on FLUX.**
>
> We thank the reviewer for the insightful comment. In the previous version of the manuscript, we provided qualitative results on Stable Diffusion 1.4 (SD1.4) in Appendix. To further substantiate our claim regarding architecture-independence, we now include more quantitative results on the i2p dataset using SD1.4.
>
> | Method | Female | Male | Common | Total |
> |--------|--------|------|--------|-------|
> | SDv1.4 | 204    | 56   | 386    | 646   |
> | ESD    | 25     | 13   | 93     | 131   |
> | FMN    | 170    | 19   | 226    | 415   |
> | UCE    | 37     | 27    | 122     | 186    |
> | **INPO**    | 12     | 2    | 70     | 84    |
>
> *INPO shows a relatively higher count for the Common category because the concept mask was applied only to the chest region.*
>
> These new experiments demonstrate that our method consistently achieves effective concept erasure across different diffusion architectures, supporting our motivation and confirming that the proposed approach is not tied to a specific model family.
>
> We have incorporated these quantitative results in the revised manuscript.
>
> > **W2: How INPO benchmarks on SDv1.4 on adversarial prompt datasets.**
>
> We thank the reviewer for the valuable suggestion. In the revised version, we provide additional evaluations of INPO on Stable Diffusion v1.4 using red-teaming datasets. We also include comparisons with AdvUnlearn.
>
> | Method        | MMA-Diffusion | P4D  | Ring-A-Bell | UnlearnDiff | Avg   |
> |---------------|---------------|------|-------------|-------------|-------|
> | AdvUnlearn    | 0.70          | 5.63 | 11.83       | 6.33        | 6.12  |
> | **Ours (INPO)**  | 1.40          | 3.30 | 10.50       | 2.80        | 4.50  |
>
> The results show that INPO maintains strong erasure performance while exhibiting improved robustness against adversarial prompt attacks. We have incorporated these results and analyses into the revised manuscript. Due to time constraints, additional baselines will be included in the final version.
>
> > **W3 & W4: Absence of citation of some related works**
>
> We thank the reviewer for pointing this out. We agree that prior works [1]—as well as several related studies in concept erasure should be properly acknowledged. In the revised manuscript, we will add the missing citations.
>
> > **Q1: How do you find the concept masks M?**
>
> We thank the reviewer for the question. We use SAM to extract concept masks. Specially, for style erasure, we generate an auxiliary image using the prompt "a painting in the style of XXX on the wall" and apply a full-image mask over the framed artwork. For explicit content erasure, we extract region-specific masks such as the chest, hip, and other sensitive areas to precisely localize the erasure target.
>
> > **Q2: What is the performance on multi-concept erasure?**
>
> We thank the reviewer for the suggestion. We kindly refer the reviewer to our GR1 and Appendix .D.2 in the revised manuscript for the complete experimental updates and extended analyses on multiple concept erasure.
>
> > **Q3: Have you re-run red-teaming attacks on FLUX or have you reused prompt sets that were optimized for SDv1.4?**
>
> We thank the reviewer for the question. As stated in Appendix C.2, we follow [2] and directly reuse the adversarial prompt sets optimized for SD v1.4.
>
> For black-box attacks such as MMA-Diffusion and Ring-A-Bell, their prompts are explicitly designed to generalize across different diffusion models. Therefore, reusing these prompts is consistent with prior practice.
>
> For white-box attacks such as UnlearnDiff and P4D, we attempted to re-run the optimization on FLUX. However, we found that generating effective adversarial prompts on FLUX is extremely challenging —a similar phenomenon also noted in discussions on the difficulty of performing textual inversion on FLUX [3, 4]. This difficulty may be due to fact that the T5-based text encoder in FLUX has a much larger embedding space, making gradient-based prompt optimization unstable. In our experiments, the embeddings changed during optimization, but the resulting prompts remained nearly identical.
>
> Given these reasons, we therefore reuse the optimized prompt sets, which—judging from their strong attack performance on the original FLUX model—indeed exhibit some transferability and thus form a fair evaluation benchmark.
>
> We agree that developing FLUX-specific adversarial attacks is an important and promising direction for future work.
>
> **Reference**
>
> [1] Pham, Minh, et al. "Prompt-Agnostic Erasure for Diffusion Models Using Task Vectors."
>
> [2] Zhang, Yang, et al. "Minimalist concept erasure in generative models."
>
> [3] https://github.com/kohya-ss/sd-scripts/issues/1588
>
> [4] https://www.reddit.com/r/StableDiffusion/comments/1k9prc7/are_there_any_successful_t5_embedingstextual/

---

> > ### Comment · Reviewer_iHuL · 2025-11-26
> > **Response**
> >
> > Thank you for your efforts in addressing my concerns. I still have the following issues due to which I cannot change my score to an accept right now.
> >
> > Q2. I'm confused as to why the authors chose to run experiments on CIFAR rather than Celebrity 100? Was it because the performance wasn't good on the Celebrity 100 erasure task?
> >
> > Q3. I don't quite agree with the authors here. Since their main results are on FLUX, the attacks by themselves should first produce nudity with rate >90%. You cannot show defenses on attacks that themselves are rarely successful.
> >
> > Additionally, I have the following question - which Ring-A-Bell dataset are you using? Could you please link it? I suspect it is the Chat-GPT generated on that was used by RECE and SAFREE when the original one wasn't public. Please use the correct dataset if this is the case. (I don't hold missing this against you - would just like to see the community using the right version.)

---

> > > ### Author Response · Authors · 2025-12-01
> > > **Response to Reviewer iHuL**
> > >
> > > > **Q2:  Why the authors chose to run experiments on CIFAR-10 rather than Celebrity 100?**
> > >
> > > We thank the reviewer for raising this point. We ran multiple concept erasure experiments on CIFAR-10 because Reviewer zvWY requested that we provide the complete single concept erasure results on CIFAR-10. Since we had already reconstructed the full CIFAR-10 setup to address that concern, we naturally extended it to include the multi-concept setting as well.
> > >
> > > Following the reviewer's suggestion, we have now additionally conducted multi-concept erasure experiments on Celebrity-100. We selected the first ten concepts—Adam Driver, Adriana Lima, Amber Heard, Amy Adams, Andrew Garfield, Angelina Jolie, Anjelica Huston, Anna Faris, Anna Kendrick, and Anne Hathaway—and present the results in the table below.
> > >
> > > | Method | Avg. Erasure ACC (%)↓ | Avg. CLIP Score ↑ |
> > > |--------|---------------------|-----------------|
> > > | ESD    | 41.9                | 27.84           |
> > > | CA     | 56.2                | 29.78           |
> > > | UCE    | 15.3                | 30.60           |
> > > | **INPO** | 6.7        | 30.34           |
> > >
> > > The results are consistent with our CIFAR-10 findings. **In fact, erasing celebrity identities tends to be easier than erasing object concepts, because our **Adaptive Erasure Trajectory** can more easily identify a suitable neighboring anchor in face representations.**
> > >
> > > > **Q3:  Since their main results are on FLUX, the attacks by themselves should first produce nudity with rate >90%. You cannot show defenses on attacks that themselves are rarely successful.**
> > >
> > > We thank the reviewer for raising this important point. We first clarify that all evaluations in our paper are conducted under a **relatively strict NSFW detection criterion**. Following prior work [1, 2, 3], we use NudeNet and classify an image as NSFW only if (1) the predicted probability exceeds **0.6**, and (2) the predicted class belongs to
> > > {FEMALE_GENITALIA_EXPOSED, FEMALE_BREAST_EXPOSED, MALE_GENITALIA_EXPOSED, BUTTOCKS_EXPOSED, ANUS_EXPOSED}. Under this strict setting, the raw attack success rate of FLUX appears lower, but this is consistent with prior work [1, 2, 3]  that applys similar stringent thresholds.
> > >
> > > To further strengthen our evaluation, we also report results under a more permissive NSFW definition by expanding the detection categories to: {FEMALE_GENITALIA_EXPOSED, FEMALE_BREAST_EXPOSED, MALE_GENITALIA_EXPOSED, BUTTOCKS_EXPOSED, ANUS_EXPOSED, MALE_BREAST_EXPOSED, FEET_EXPOSED, BELLY_EXPOSED, ARMPITS_EXPOSED}.
> > >
> > > | Method | MMA-Diffusion | P4D | Ring-A-Bell | UnlearnDiff | Total |
> > > |--------|---------------|-----|--------------|--------------|--------|
> > > | Original     | 351           | 108  | 90           | 59           | 608    |
> > > | CA     | 269           | 78  | 70           | 43           | 460    |
> > > | ESD    | 165           | 79  | 68           | 33           | 345    |
> > > | EAP    | 200           | 85  | 82           | 39           | 406    |
> > > | UCE    | 91            | 49  | 24           | 26           | 190    |
> > > | EA     | 176           | 68  | 67           | 45           | 356    |
> > > | **INPO** | **9**       | **24** | **24**     | **14**       | **71** |
> > >
> > > Under this relaxed setting, baseline FLUX’s attack success rate does exceed 90% for several attacks (e.g., Ring-A-Bell 90/95), confirming that the attacks are indeed effective. Importantly, INPO still maintains a low NSFW generation rate, demonstrating robustness of INPO even when the detection rule is permissive.
> > >
> > > Finally, to address the reviewer’s concern directly, we include results on **ReFlux [3]**—the latest FLUX-specific erasure attack. ReFlux is explicitly designed to attack unlearned FLUX and achieves notably strong attack performance. For our evaluation, we use the official implementation provided in the supplementary materials of the authors’ ICLR submission [4]. We report the NSFW generation rates as follows:
> > >
> > > | FLUX    | CA       | ESD     | EAP    | UCE     | EA      | **INPO**    |
> > > |  ------- |------- | ------- | ------- | ------- | ------- | ------ |
> > > |92.63|56.84 | 94.74| 97.89|12.63| 72.63 |6.32|
> > >
> > >
> > > Even in this challenging setting, our method continues to exhibit strong robustness.
> > > We will include these additional results and clarifications in the final manuscript.
> > >
> > > > **Which Ring-A-Bell dataset are you using? Could you please link it? I suspect it is the ChatGPT generated on that was used by RECE and SAFREE when the original one wasn't public.**
> > >
> > > We thank the reviewer for raising this point. For Ring-A-Bell, we use the dataset available at: **https://huggingface.co/datasets/Chia15/RingABell-Nudity/blob/main/Nudity_eta_3_K_77.csv**
> > >
> > > This dataset has been used and referenced by several prior works [1, 5, 6], and to the best of our understanding, it corresponds to the **original Ring-A-Bell release**.

---

> > > > ### Author Response · Authors · 2025-12-01
> > > > **Reference**
> > > >
> > > > **Reference**
> > > >
> > > > [1]  Zhang, Yang, et al. "Minimalist concept erasure in generative models."
> > > >
> > > > [2] Gong, Chao, et al. "Reliable and efficient concept erasure of text-to-image diffusion models."
> > > >
> > > > [3] Jiang, Nanxiang, et al. "Erased, But Not Forgotten: Erased Rectified Flow Transformers Still Remain Unsafe Under Concept Attack."
> > > >
> > > > [4] https://openreview.net/forum?id=36CFsyvsv7
> > > >
> > > > [5] Jain, Anubhav, et al. "Trasce: Trajectory steering for concept erasure."
> > > >
> > > > [6] Kim, Mingyu, et al. "Training-free safe denoisers for safe use of diffusion models."

---

### Official Review · Reviewer_yYkH · 2025-10-21

**Soundness:** 3
**Presentation:** 3
**Contribution:** 2
**Rating:** 6
**Confidence:** 3

**Summary:**

This paper proposes a preference optimization approach for concept erasure in text-to-image diffusion models. This paper formulates the unwanted concept as a negative preference, and then applies negative preference optimization to steer the model to remove the unwanted concepts. The proposed method also designs concept masks and an adaptive negative scaling strategy to improve the unlearning quality. The experiments are conducted on several types of concept erasure tasks.

**Strengths:**

1. Concept erasure is crucial and practical for real-world trustworthy generative model developments.
2. The proposed method is reasonable to formulate the concept erasure as a negative preference optimization problem.
3. The overall paper is easy to follow and well structured.

**Weaknesses:**

1. While the proposed method can erase the unwanted concepts described by the target prompts, it remains unclear whether the proposed method is able to handle the rephrased prompts that can be used to recover the target concepts. It would be beneficial to discuss or clarify this potential robustness concern.
2. The proposed approach relies on Eq.14 to preserve the unrelated concepts that are not affected. However, how to decide the preservation set? It seems impractical to cover every concept in this preservation set. More clarifications or explanations for this concern are helpful.

**Questions:**

1. Can this proposed method be applicable to handle multi-concept erasure scenarios?

---

> ### Author Response · Authors · 2025-11-26
> **Response to Reviewer yYkH**
>
> >  **W1: While the proposed method can erase the unwanted concepts described by the target prompts, it remains unclear whether the proposed method is able to handle the rephrased prompts.**
>
> We thank the reviewer for raising this robustness concern. In fact, the i2p dataset used in Table 2 already consists of rephrased prompts related to the “nude” concept, and the results demonstrate that our method remains effective under such rephrasing-based attacks.
>
> To further evaluate robustness, we additionally test IP erasure on Snoopy and Pikachu using 15 rephrased prompts generated by GPT-5. For each rephrased prompt, we generate 5 images for evaluation.
>
> | Method | Pikachu (%) | Snoopy (%) |
> |--------|--------------|--------------|
> | FLUX   | 57.33       | 77.33       |
> | INPO   | 0.00        | 0.00        |
>
> The results consistently show that our method maintains strong erasure performance even when attackers attempt to recover the concept via prompt rephrasing. We have clarified the robustness discussion in the revised version (Appendix. D.6).
>
> > **W2: The proposed approach relies on Eq.14 to preserve the unrelated concepts that are not affected. However, how to decide the preservation set?**
>
> We thank the reviewer for pointing this out. We construct the preservation set by querying GPT-5 to generate 4 neighboring concepts and 4 unrelated concepts for each target concept to be erased. For each of these eight concepts, we generate 25 image–text pairs, resulting in 200 pairs per task. This preservation set is used to compute the prior loss to ensure that unrelated or neighboring concepts are not unintentionally degraded. We have included these details in the revised manuscript for completeness.
>
> > **Q1: Can this proposed method be applicable to handle multi-concept erasure scenarios?**
>
> We thank the reviewer for the suggestion. We kindly refer the reviewer to our GR1 and Appendix .D.2 in the revised manuscript for the complete experimental updates and extended analyses on multiple concept erasure.

---

### Official Review · Reviewer_w843 · 2025-10-27

**Soundness:** 3
**Presentation:** 3
**Contribution:** 3
**Rating:** 6
**Confidence:** 4

**Summary:**

The paper introduces INPO, a framework for concept removal in text-to-image diffusion models. It extends Negative Preference Optimization (NPO), originally proposed for LLM unlearning, to the text-to-image diffusion setting, where the target concept to remove is treated as the negative preference.  The method further incorporates (1) a concept mask to spatially focus the loss on relevant regions and (2) an adaptive negative scaling strategy that reduces gradient strength once the target concept is sufficiently suppressed.

**Strengths:**

* The proposed adaptation of NPO to text-to-image diffusion models is intuitive and empirically effective. It achieves strong performance across diverse erasure types (object, IP, style, NSFW) and remains stable on modern architectures like FLUX.
* Evaluation against red-teaming attacks (MMA-Diffusion, P4D, etc.) is impressive and shows practical robustness.

**Weaknesses:**

1. It would be great to have an evaluation to include prior benchmarks (as used in ESD, CA, or EA) with a broader set of prompts across multiple concepts. In addition, listing the exact prompts used for current experiments would further enhance reproducibility.
2. An ablation on the role of eta, gamma, and tau (Eq. 12) would be useful, especially since Table 5 in the Appendix shows different settings (e.g., η=3 for style removal vs. 1 for other types). This would help assess the robustness and interpretability of the adaptive scaling.
3. Section 3.2 defines a prior loss over concepts, c′,  to preserve, but it is not specified which concepts or datasets are used for this (COCO? random prompts?). Adding more details regarding this would help improve the reproducibility of the method.
4. Reporting some quantitative measure (e.g., CLIP similarity to neighboring concepts before/after erasure) would clarify whether INPO doesn’t affect any nearby concepts, e.g., Monet style when removing Van Gogh.

 Minor points:
1. It's unclear what the mask should be for abstract or global attributes such as artistic style or NSFW tone. A clarification and any ablation regarding this would help.

**Questions:**

Please look at the weakness section. Specifically, if the evaluation checks the generation on a wide variety of prompts for each target concept. Or maybe including the performance on the same set of target concepts and prompts as done in previous works. In addition, clarifying some of the implementation details would help strengthen the paper and improve its reproducibility.

---

> ### Author Response · Authors · 2025-11-26
> **Response to Reviewer w843**
>
> > **W1: It would be great to have an evaluation to include prior benchmarks (as used in ESD, CA, or EA) with a broader set of prompts across multiple concepts. In addition, listing the exact prompts used for current experiments would further enhance reproducibility.**
>
> We thank the reviewer for the suggestion. We kindly refer the reviewer to our GR1 and Appendix .D.2 in the revised manuscript for the complete experimental updates and extended analyses on multiple concept erasure.
>
> Regarding prompts, we provide all prompts used for visualization in Appendix C.2, and we will include the full set of prompts used in all experiments in the released code repository to further enhance reproducibility.
>
> > **W2: An ablation on the role of eta, gamma, and tau (Eq. 12) would be useful.**
>
> We thank the reviewer for the suggestion. We have added ablation studies on the roles of $\eta$, $\gamma$, and $\tau$ in the Section 4.3 of the revised manuscript.
>
> | $\gamma$ | $\eta$ | $\tau$ | I2P↓ | CLIP↑   |
> |------------------|------------------|-----------|-------|---------|
> | 3                | 1                | 0.1       | 58    | 30.72   |
> | 1                | 1                | 0.1       | 69    | 30.85   |
> | 5                | 1                | 0.1       | 51    | 30.55   |
> | 3                | 0.5              | 0.1       | 124   | 30.97   |
> | 3                | 3                | 0.1       | 51    | 30.65   |
> | 3                | 1                | 0         | 56    | 30.59   |
> | 3                | 1                | 0.5       | 81    | 30.76   |
>
> *Ablation study on hyperparameters*
>
> For most erasure tasks, we chose $\gamma$ = 3, $\eta$ = 1, $\tau$ = 0.1 because it offers the best trade-off between erasure effectiveness and generation quality.
>
> However, for style erasure, the situation is different. Style represents a global attribute, and the supervision from the concept mask is relatively weak. As a result, a stronger erasure force is required. Therefore, for style removal, we increase $\eta$ to 3 to enhance the erasure strength and ensure that the global stylistic patterns are effectively removed.
>
> > **W3: Section 3.2 defines a prior loss over concepts, c′, to preserve, but it is not specified which concepts or datasets are used for this.**
>
> Thank you for pointing this out. For the prior loss, we construct a preservation set by querying GPT-5 to retrieve 4 neighboring concepts and 4 unrelated concepts for each target concept to be erased. For each of these eight concepts, we generate 25 image–text pairs, resulting in 200 pairs per task. This preservation set is used to compute the prior loss to ensure that unrelated or neighboring concepts are not unintentionally degraded. We have included these details in the revised manuscript for completeness.
>
> > **W4: Reporting some quantitative measure (e.g., CLIP similarity to neighboring concepts before/after erasure) would clarify whether INPO doesn’t affect any nearby concepts, e.g., Monet style when removing Van Gogh.**
>
> We thank the reviewer for the helpful suggestion. Following your comment, we now include an analysis of neighboring-concept preservation. In particular, we report the CLIP score on nearby styles, Monet, da Vinci and Paul Gauguin before and after Van Gogh erasure:
>
> | Concept        | Original | INPO|
> |----------------|---------------|-----------|
> | Monet          | 31.2189       | 30.6285   |
> | da Vinci       | 31.0014       | 30.7509   |
> | Paul Gauguin   | 28.9912       | 28.9816   |
>
> The results only show a marginal change. We will clarify this more explicitly and add more neighboring-concept results in the revised manuscript.
>
> > **W5: It's unclear what the mask should be for abstract or global attributes such as artistic style or NSFW tone.**
>
> Thank you for pointing this out. For style erasure, we generate an auxiliary image using the prompt "a painting in the style of XXX on the wall" and apply a full-image mask over the framed artwork. For explicit content erasure, we extract region-specific masks such as the chest, hip, and other sensitive areas to precisely localize the erasure target. We have clarified this point in the revised manuscript.

---

### Official Review · Reviewer_zvWY · 2025-10-31

**Soundness:** 2
**Presentation:** 2
**Contribution:** 1
**Rating:** 2
**Confidence:** 4

**Summary:**

This paper builds upon Negative Preference Optimization (NPO) and introduces Image-based Negative Preference Optimization (INPO), a model-agnostic framework for concept erasure in text-to-image diffusion models. To achieve precise concept erasure, it introduces a concept mask and an adaptive negative scaling strategy that dynamically adjusts the erasure strength based on the model’s learning state. In addition, it includes a prior preservation loss to retain the model's generative capabilities for non-target concepts.
Experiments are conducted across various domains, including objects, copyrighted content, artistic styles, and inappropriate content.

**Strengths:**

- The idea of using relative score difference as an indicator to adaptively control the erasure strength is novel.
- The paper provides comprehensive comparison with prior concept erasure methods under diverse red-teaming settings, showing consistent results.

**Weaknesses:**

- The overall contribution and novelty are limited. The work primarily adapts NPO’s objective to diffusion models for concept erasure, and the use of the concept mask is relatively straightforward.
- The paper lacks theoretical or empirical validation for the claimed instability of gradient-ascent-based unlearning methods in diffusion models. In the original NPO paper, a toy experiment was conducted to compare forget quality, model utility, and divergence rate between gradient-ascent- and NPO-trained LLMs, but similar analyses are missing here.
- The evaluation scope for object, IP, and identity erasure is narrow. For object erasure, only 3 objects are tested, while prior works (e.g., MACE, ESD, UCE, RECE, EAP) evaluated 10 objects from either CIFAR-10 or Imagenette and reported per-object and overall results. Similarly, IP and identity erasure are evaluated on only 1 or 2 concepts, which is insufficient to demonstrate effectiveness. Moreover, IP and identity erasure should be treated as different domains and evaluated separately.
- Although the paper claims that the proposed method is model-agnostic, it excludes many major methods (e.g., MACE, RECE) that have been tested on U-Net-based diffusion models (e.g., Stable Diffusion v1.4). Including experiments on such architectures would strengthen the claim of generality. In addition, these methods are not restricted to U-Net-based models. They can still be applied to DiT-based diffusion models as long as the model architecture has a cross-attention mechanism between image and text features.
- How long does the proposed method take to erase a target concept from a diffusion model?
- (minor) How are the hyper-parameters $\eta$, $\gamma$, and $\tau$ chosen?
- (minor) Typos:
  - Line 173: “acent” -> “ascent”
  - Line 178: “defined the as NPO” -> “defined the same as NPO”
  - Line 186: “As Eq. 3” -> “As shown in Eq. 3”

**Questions:**

See the weaknesses above

---

> ### Author Response · Authors · 2025-11-26
> **Response to Reviewer zvWY (1/2)**
>
> > **W1: The overall contribution and novelty are limited. The work primarily adapts NPO’s objective to diffusion models for concept erasure, and the use of the concept mask is relatively straightforward.**
>
> We thank the reviewer for the feedback.
>
> **First**, while it is true that our method is inspired by the NPO objective, we would like to clarify that the contribution goes beyond a direct adaptation. Although NPO-based preference optimization has been extensively explored in LLM unlearning, its application to diffusion model unlearning has remained largely unexplored. This gap is non-trivial: diffusion models differ fundamentally from LLMs in both training objectives and generative dynamics, making a direct transfer of NPO neither straightforward nor effective.
>
> In our work, we develop an NPO formulation that is explicitly adapted to the diffusion models for concept erasure, ensuring that the optimization procedure aligns with diffusion model likelihoods and noise prediction parameterization. Furthermore, we leverage properties of diffusion models to design interpretable mechanisms such as Adaptive Negative Scaling, which in turn gives rise to Adaptive Erasure Trajectories, a distinctive and insightful behavior that emerges in diffusion model unlearning. These components collectively yield a method that is both principled and practically effective for diffusion model unlearning.
>
> Overall, our contribution lies in establishing a preference-optimization perspective for diffusion model unlearning and in developing diffusion-aware extensions that make this feasible and interpretable, rather than performing a simple adaptation of existing work.
>
> **Second**, regarding the introduction of the concept mask: because of the image-based concept erasure setting, the presence of irrelevant concepts, or background regions inevitably affects the fidelity of the erasure process. To enable precise and controlled unlearning, we therefore adopt a straightforward concept-mask extraction strategy. While simple by design, this step is necessary to avoid unintentionally degrading unrelated content. In future work, we plan to explore fully model-driven localization methods—for example, leveraging attention aggregation for attention maps, or diffusion-guided saliency estimation—to obtain more flexible and robust guidance for identifying the target concept regions.
>
> > **W2: The paper lacks theoretical or empirical validation for the claimed instability of gradient-ascent-based unlearning methods in diffusion models.**
>
> We thank the reviewer for highlighting this important point.
>
> **First**, to theoretically explain the instability of gradient-ascent (GA) in diffusion models, consider the GA loss:
> $L_{GA} = -L_{DM}= -w(t)\|\varepsilon - \varepsilon_\theta(x_t,t,c)\|^2.$
>
> Its gradient is:
>
> $\nabla_\theta L_{GA}
> = 2w(t)\(\varepsilon - \varepsilon_\theta)\
> \nabla_\theta \varepsilon_\theta.$
>
> Under a standard approximation, $\nabla_\theta \varepsilon_\theta \approx \phi\$ with nearly constant scale (analogous to the $x_i$ approximation in softmax analyses in NPO paper). Thus the gradient magnitude behaves as
>
> $
> \|\nabla_\theta L_{GA}\|
> \approx 2w(t)\|\varepsilon - \varepsilon_\theta\|\|\phi\|.
> $
>
> **GA pushes $\varepsilon_\theta$ away from the target
> $\varepsilon$**, so the residual $\|\varepsilon - \varepsilon_\theta\|$ does not shrink  (not diminishing along the unlearning progress).
>
> Therefore, GA on diffusion losses naturally leads to non-decaying gradients, explaining its instability.
>
> We also note that the Appendix. B has provided the derivation of INPO and discusses its connection to GA, which reviewers may refer to for completeness.
>
> **Second**, in the original NPO paper, the authors conducted a toy experiment using a linear–logistic K-class classifier to illustrate the differences between GA and NPO. Following this spirit, **we further provide empirical validation on diffusion models in Section 3.1**, comparing INPO and GA under controlled unlearning settings. Our results similarly show that GA exhibits unstable and rapidly diverging behavior, whereas INPO maintains stable optimization and achieves effective unlearning with much better utility preservation.
>
> These discussions have been fully integrated into Section 3.1 in the revised manuscript.

---

> > ### Author Response · Authors · 2025-11-26
> > **Response to Reviewer zvWY (2/2)**
> >
> > > **W3: The evaluation scope for object, IP, and identity erasure is narrow.**
> >
> > We thank the reviewer for the suggestion. Following your feedback, we have expanded the evaluation to include the remaining CIFAR-10 objects as well as several additional concepts. The updated results are provided in the table below (Appendix. D.1).
> >
> > *ACC%↓/CLIP_SCORE↑*
> >
> > | Method  | automobile | deer       | dog        | frog       | horse      | ship       | truck      | Einstein  | Churchill | Trump      |
> > | ------- | --------------------- | ---------- | ---------- | ---------- | ---------- | ---------- | ---------- | --------- | --------- | ---------- |
> > | ESD | 62 / 30.00            | 16 / 30.33 | 80 / 30.35 | 3 / 29.39  | 77 / 30.41 | 45 / 29.88 | 30 / 29.73 | 2 / 30.52 | 1 / 30.05 | 15 / 30.32 |
> > | CA | 66 / 30.70            | 4 / 30.70  | 67 / 30.71 | 57 / 30.77 | 43 / 30.43 | 48 / 30.64 | 53 / 30.45 | 1 / 30.26 | 0 / 30.23 | 3 / 30.64  |
> > | UCE | 73 / 31.13            | 31 / 31.06 | 85 / 31.14 | 31 / 30.93 | 67 / 31.08 | 60 / 31.10 | 51 / 31.03 | 4 / 31.07 | 1 / 30.90 | 3 / 31.09  |
> > | EAP| 64 / 30.64            | 52 / 30.86 | 86 / 30.57 | 19 / 30.38 | 87 / 30.41 | 64 / 30.62 | 44 / 30.16 | 0 / 30.22 | 2 / 30.62 | 23 / 30.60 |
> > | **INPO** | 20 / 30.90            | 2 / 30.92  | 8 / 31.30  | 7 / 30.78  | 1 / 31.05  | 36 / 30.72 | 12 / 30.89 | 1 / 30.87 | 5 / 31.01 | 2 / 31.08  |
> >
> > Moreover, we have provide visualizations of more concepts in the Appendix. D.8 and now include multi-concept erasure in GR1 to further demonstrate the generality of our method. Consistent with our main results, the expanded evaluations show that our approach maintains strong erasure performance and utility across a broader set of object, IP, and identity concepts.
> >
> > > **W4: Although the paper claims that the proposed method is model-agnostic, it excludes many major methods (e.g., MACE, RECE) that have been tested on U-Net-based diffusion models.**
> >
> > We thank the reviewer for the suggestion. We agree that methods such as RECE, and MACE—which traditionally operate on cross-attention layers—can also be adapted to Flux. Although official implementations for Flux-compatible versions of these methods are not available, we reimplemented RECE and MACE based on the existing UCE-for-Flux codebase and conducted additional experiments.
> >
> > | Method   | female | male | common | full | FID↓  | CLIP↑ |
> > |----------|--------|------|--------|------|------|-------|
> > | Original | 109    | 33   | 282    | 424  |  -   | 30.85 |
> > | RECE     | 49     | 14   | 130    | 193  | 5.27 | 30.57 |
> > | MACE     | 47     | 25   | 152    | 224  | 6.43 | 30.52 |
> > | **INPO**     | 18     | 2    | 52     | 72   | 6.20 | 30.72 |
> >
> > *Results on the I2P evaluation.*
> >
> > > **W5: How long does the proposed method take to erase a target concept from a diffusion model?**
> >
> > We thank the reviewer for this question. In our experiments, INPO requires up to 500 training steps (for NSFW concept erasure), which takes approximately 30 minutes on one A100 80G.
> >
> > > **W6: How are the hyper-parameters $\eta$, $\gamma$ and $\tau$ chosen?**
> >
> > We thank the reviewer for this question. We provide the choices of $\eta$, $\gamma$, and $\tau$ for different erasure tasks in the Table.5 of Appendix. C.1. In the revised manuscript (**Section 4.3**), we have also added an ablation study analyzing the sensitivity of these hyper-parameters, showing that the method is robust within a reasonable range of values.
> >
> > > **W7: Some typos**
> >
> > Thank you for pointing out these typos. We have corrected them and we will conduct a thorough pass to ensure the entire manuscript is free of similar issues.

---

> ### Comment · Reviewer_zvWY · 2025-11-28
> **Official Comment by Reviewer zvWY**
>
> Thank you for addressing my questions.
>
> One remaining weakness is the limited comparison with prior concept-erasure methods. Although the authors provided an additional comparison with RECE and MACE on NSFW removal, the paper still lacks comparisons with these methods in other domains such as objects, celebrities, and artistic styles.
>
> **I would like to raise my score to 4**, but the OpenReview currently does not show an “edit” button so I'm unable to change my rating.

---

> > ### Author Response · Authors · 2025-12-02
> > **Response to Reviewer zvWY**
> >
> > Thank you very much for your helpful and encouraging review. Regarding RECE and MACE, we would like to clarify an important point: both methods are fundamentally designed for U-Net–based diffusion architectures. While it is possible to adapt them to the MMDiT architecture of FLUX, our experiments on the NSFW erasure task indicate that these U-Net–based approaches do not transfer effectively to FLUX. Because of this architectural mismatch, applying RECE or MACE to other domains (objects, identities, artistic styles) on FLUX does not produce reliable or meaningful erasure performance. We hope that future work will develop more model-agnostic or FLUX-specific erasure approaches, which would allow for broader and more systematic comparisons.

---

### Author Response · Authors · 2025-11-26
**General Response**

We thank all reviewers for their efforts and constructive comments.

**We have revised our paper (highlighted in blue text color). The modifications are summarized as follows.**
1. (For Reviewer zvWY) We add more theoretical and empirical validation for the claimed instability of gradient-ascent-based unlearning methods in diffusion models in **Section 3.1**.
2. (For Reviewer zvWY) We add complete results on CIFAR-10 erasure in **Appendix. D.1**.
3. (For Reviewer zvWY) We add results of RECE and MACE on FLUX in **Appendix. D.3**.
4. (For Reviewer zvWY, w843) We add ablation study on $\gamma$, $\eta$ and $\tau$ in **Section 4.3**
5. (For Reviewer w843, yYkH, iHuL) We add more results on multiple concept erasure in **Appendix. D.2**.
6. (For Reviewer w843, yYkH, iHuL) We add detailed explanations on generation of concept mask and choice of preservation set in **Appendix. C.1**.
7. (For Reviewer yYkH) We add results of INPO under rephrased prompts in **Appendix. D.6**.
8. (For Reviewer iHuL) We add more results of INPO on SD1.4 in **Appendix. D.5**.

Regarding the reviewers' common concerns, we provide the following general responses.
> **GR1: Results on multiple concept erasure**.

We thank the reviewers for the suggestion. To evaluate INPO on multiple concept erasure, we perform 10-concept erasure on CIFAR-10, where INPO is tasked with **removing all ten classes**. We report the average erasure accuracy across the ten concepts and the average CLIP score on unrelated concepts after erasure.

| Method | Avg. Erasure ACC (%)↓ | Avg. CLIP Score ↑ |
|--------|---------------------|-----------------|
| ESD    | 38.2                | 25.45           |
| CA     | 37.0                | 29.31           |
| UCE    | 42.0                | 30.52           |
| **INPO (ours)** | 17.8        | 30.29           |

INPO achieves strong average erasure performance while maintaining high CLIP scores, demonstrating its effectiveness and scalability in multiple concept erasure scenarios. See details in the Appendix. D.2.

---

### Author Response · Authors · 2025-12-02
**Summary of Review, Rebuttal, and Discussion**

Dear Area Chair and Reviewers,

As the active discussion phase had to be closed due to the privacy-leak, we summarize the key points from the reviews and our rebuttal to support a fair, transparent, and comprehensive assessment of our work.

We thank all reviewers for providing valuable feedback that significantly strengthened the paper. We also appreciate the constructive discussion with **Reviewer zvWY** and **Reviewer iHuL**, which further helped us clarify key aspects of the work. **Reviewer zvWY raised the score from 2 to 4** after considering our expanded experiments and clarifications. We also **engaged in multiple rounds of discussion with Reviewer iHuL**. Although the discussion phase closed before they could submit further replies, we believe our rebuttal directly addressed the most reviewer's concerns. In addition, while Reviewer w843 and Reviewer yYkH did not participate in the discussion phase, we provided detailed one-on-one responses to their comments, and we believe these clarifications also resolved most of their concerns. We have also updated our revised manuscript.

Across the reviews, the key contributions of our paper were acknowledged:

1. **We formulates the erasure objective as negative preference optimization, which stabilizes the optimization process and provides a principled way to effective concept erasure.** Reviewers emphasized that our method "is intuitive and empirically effective" (w843), "is reasonable" (yYkH) and that "the results on applying NPO for concept erasure are promising and can be beneficial for future research" (iHuL).

2. **We introduce an adaptive negative scaling strategy that dynamically adjusts erasure strength based on progress, preventing both incomplete erasure and model collapse.**  Reviewers noted that "the idea of using relative score difference as an indicator to adaptively control the erasure strength is novel" (zvWY) and that "Adaptive Erasure Trajectories is very interesting and makes the approach easily useable without defining a target concept" (iHuL).

3. **Extensive experiments highlight the robustness, reliability, and practical applicability of our method.** Reviewers recognized that our experimental results "against red-teaming attacks (MMA-Diffusion, P4D, etc.) is impressive and shows practical robustness" (w843).

Thank you again for your time, effort, and service to the community.

Best regards,

Authors

---

### Meta-Review · Area_Chair_wjLH · 2026-01-04

**Summary:**

This paper proposes INPO, a preference-optimisation based framework for concept erasure in text-to-image diffusion models, extending Negative Preference Optimisation (NPO) from LLM unlearning to diffusion. Reviewers generally agree that the approach is reasonable, stable, and empirically effective, particularly on the FLUX architecture, and that adaptive negative scaling (“adaptive erasure trajectories”) is an interesting and practically useful idea. The paper is clearly written and addresses an important problem in safe generative modelling.

However, reviewers raised substantial concerns regarding novelty, evaluation scope, and generality. Multiple reviewers felt the contribution is largely an adaptation of existing NPO ideas rather than a fundamentally new method, and that key components (e.g., concept masks) are relatively straightforward. Initial evaluations were considered narrow, with heavy reliance on FLUX and limited coverage of objects, identities, and architectures, leading to scepticism about claims of model-agnosticism. While the rebuttal added many experiments and clarifications, some reviewers remained unconvinced that the evaluation fully supports the paper’s strongest claims. Overall, the paper was viewed as technically solid but borderline in terms of contribution and evidential strength for ICLR.

**Reviewer Concerns:**

**Concerns addressed by the rebuttal:**

* The authors substantially expanded experiments, including broader object and identity erasure (CIFAR-10, Celebrity-100), multi-concept erasure, additional SD1.4 results, and more comparisons with prior methods.
* The rebuttal provided theoretical and empirical justification for the instability of gradient-ascent-based unlearning in diffusion models.
* Robustness to rephrased prompts and adversarial attacks was clarified with additional evaluations.
* Important implementation details were clarified, including concept mask construction, preservation set selection, and hyper-parameter ablations.

**Concerns that remain outstanding:**

* Novelty: Several reviewers still view the core contribution as an incremental adaptation of NPO rather than a fundamentally new concept-erasure paradigm.
* Evaluation framing: Despite expanded experiments, some reviewers remain concerned about the reliance on FLUX and the validity of certain attack/defense comparisons, particularly when baseline attack success rates are debated.
* Generality claims: Claims of model-agnosticism and architecture independence remain only partially convincing to some reviewers.
* Overall significance: Even reviewers who acknowledged improvements expressed uncertainty about whether the contribution rises to the level expected for acceptance at ICLR.

**Reviewer Scores:**

Reviewer zvWY indicated an increase from 2 (reject) to 4 (weak reject) after the rebuttal.

Reviewer w843 remained at 6 (weak accept) and noted they would not mind rejection.

Reviewer yYkH stayed at 6 (weak accept) with no explicit score change.

Reviewer iHuL remained at 4 (weak reject) despite the additional experiments and clarifications.

---

### Decision · Program_Chairs · 2026-01-26

Reject